# Out-of-distribution generalization for learning quantum dynamics

**Matthias C. Caro** [1,2,3,4,13] ✉, **Hsin-Yuan Huang** [4,5,13], **Nicholas Ezzell** [6,7], **Joe Gibbs**[8,9], **Andrew T. Sornborger**[6], **Lukasz Cincio**[10], **Patrick J. Coles**[10,11] & **Zoë Holmes**[6,12]

Generalization bounds are a critical tool to assess the training data requirements of Quantum Machine Learning (QML). Recent work has established guarantees for in-distribution generalization of quantum neural networks (QNNs), where training and testing data are drawn from the same data distribution. However, there are currently no results on out-of-distribution generalization in QML, where we require a trained model to perform well even on data drawn from a different distribution to the training distribution. Here, we prove out-of-distribution generalization for the task of learning an unknown unitary. In particular, we show that one can learn the action of a unitary on entangled states having trained only product states. Since product states can be prepared using only single-qubit gates, this advances the prospects of learning quantum dynamics on near term quantum hardware, and further opens up new methods for both the classical and quantum compilation of quantum circuits.

In quantum machine learning (QML), a quantum neural network (QNN) is trained using classical or quantum data, with the goal of learning how to make accurate predictions on unseen data[1–3]. This ability to extrapolate from training data to unseen data is known as generalization. There is much excitement currently about the potential of such QML methods to outperform classical methods for a range of learning tasks[4–11]. However, to achieve this, it is critical that the training data required for successful generalization can be produced efficiently.

While recent work has established a number of fundamental bounds on the amount of training data required for successful generalization in QML[11–24], less attention has been paid so far to the type of training data required for generalization. In particular, prior work has established guarantees for the *in-distribution generalization* of QML models, where training and testing data are assumed to be drawn

independently from the same data distribution. However, in practice one may only have access to a limited type of training data, and yet be interested in making accurate predictions for a wider class of inputs. This is particularly an issue in the noisy intermediate-scale quantum (NISQ) era[25], when deep quantum circuits cannot be reliably executed, effectively limiting the quantum training data states that can be prepared.

In this article, we study *out-of-distribution generalization* in QML. That is, we investigate generalization performance when the testing and training distributions do not coincide. Specifically, we consider the task of learning unitary dynamics, which is a fundamental primitive for a range of QML algorithms. At its simplest, the target unitary could be the unknown dynamics of an experimental quantum system. For this case, which has close links with quantum sensing[26] and Hamiltonian learning[27–29], the aim is essentially to learn a digitalization of an analog

[1]Department of Mathematics, Technical University of Munich, Garching, Germany. [2]Munich Center for Quantum Science and Technology (MCQST), Munich, Germany. [3]Dahlem Center for Complex Quantum Systems, Freie Universität Berlin, Berlin, Germany. [4]Institute for Quantum Information and Matter, Caltech, Pasadena, CA, USA. [5]Department of Computing and Mathematical Sciences, Caltech, Pasadena, CA, USA. [6]Information Sciences, Los Alamos National Laboratory, Los Alamos, NM, USA. [7]Department of Physics & Astronomy, University of Southern California, Los Angeles, CA, USA. [8]Department of Physics, University of Surrey, Guildford GU2 7XH, UK. [9]AWE, Aldermaston, Reading RG7 4PR, UK. [10]Theoretical Division, Los Alamos National Laboratory, Los Alamos, NM, USA. [11]Normal Computing Corporation, New York, NY, USA. [12]Institute of Physics, Ecole Polytechnique Fédéderale de Lausanne (EPFL), CH-1015 Lausanne, Switzerland. [13]These authors contributed equally: Matthias C. Caro, Hsin-Yuan Huang. ✉e-mail: matthias.caro@fu-berlin.de

quantum process. This could be performed using a 'standard' quantum computer or a simpler experimental system with perhaps a limited gate set, as sketched in Fig. 1a, b, respectively. Alternatively, the target unitary could take the form of a known gate sequence that one seeks to compile into a shorter depth circuit or a particular structured form[30–33]. The compilation could be performed either on a quantum computer, see Fig. 1c, or entirely classically, see Fig. 1d. Such a subroutine can be used to reduce the resources required to implement larger scale quantum algorithms including those for dynamical simulations[34–37].

Here we prove out-of-distribution generalization for unitary learning with a broad class of training and testing distributions. Specifically, we show that the average prediction error over any two *locally scrambled*[38,39] ensembles of states are perfectly correlated up to a small constant factor. This is captured by our main theorem, Theorem 1. By combining this observation with in-distribution generalization guarantees it follows that if the training and testing distributions are both locally scrambled (but potentially otherwise different distributions), out-of-distribution generalization is always possible between locally scrambled distributions. In particular, we show that a QNN trained on quantum data capturing the action of an efficiently implementable target unitary on a polynomial number of random product states, generalizes to test data composed of fully random states. That is, rather intriguingly, we show that one can learn the action of such a unitary on a broad spread of highly entangled states having only studied its action on a limited number of product states.

We numerically illustrate these analytical results by showing that the short time evolution of a Heisenberg spin chain can be well learned using only product state training data. Namely, we find that the out-of-distribution generalization error nearly perfectly correlates with the in-distribution generalization error and the training cost. In particular, in our numerical experiments, the testing performances achieved by the QML model on Haar-random states and on random product states differ only by a small constant factor, as predicted analytically. We further perform noisy simulations that demonstrate how the noise accumulated preparing highly entangled states can prohibit training. In contrast, noisy training on product states, which can be prepared using only single-qubit gates, remains feasible. Additionally, in Supplementary Note 3 we numerically validate our generalization guarantees in a task of learning so-called fast scrambler unitaries[40]. Thus our results make the possibility of using QML to learn unitary processes nearer term. Our results further suggest a new quantum-inspired classical approach to unitary compilation. Namely, our results imply that a low-entangling unitary can be compiled using only low-entangled training states. Such circuits can be readily simulated using classical tensor network methods, and hence this compilation can be performed classically.

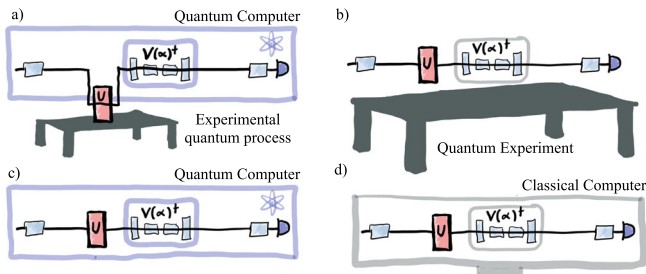

**Fig. 1 | Applications of quantum dynamics learning. a** Quantum dynamics learning of an experimental process using a quantum computer. **b** Quantum dynamics learning with a more specialized experimental system with potentially a limited gate set. **c**, **d** Quantum compilation of a known unitary on a quantum computer and classical computer, respectively.

## Results

### Framework

In this work, we consider the QML task of learning an unknown $n$-qubit unitary $U \in \mathcal{U}((\mathbb{C}^2)^{\otimes n})$. The goal is to use training states to optimize the classical parameters $\boldsymbol{\alpha}$ of $V(\boldsymbol{\alpha})$, an $n$-qubit unitary QNN (or classical representation of a QNN), such that, for the optimized parameters $\boldsymbol{\alpha}_{\mathrm{opt}}$, $V(\boldsymbol{\alpha}_{\mathrm{opt}})$ well predicts the action of $U$ on previously unseen test states.

To formalize this notion of learning, we employ the framework of statistical learning theory[41,42]. The prediction performance of the trained QNN $V(\boldsymbol{\alpha}_{\mathrm{opt}})$ can be quantified in terms of the average distance between the output state predicted by $V(\boldsymbol{\alpha}_{\mathrm{opt}})$ and the true output state determined by $U$. The average is taken over input states from a testing ensemble, which represents the ensemble of states that one wants to be able to predict the action of the target unitary on. More precisely, the goal is to minimize an *expected risk*

$$R_{\mathcal{P}}(\boldsymbol{\alpha}) = \frac{1}{4} \mathbb{E}_{|\Psi\rangle \sim \mathcal{P}} \left[ || U|\Psi\rangle\langle\Psi|U^\dagger - V(\boldsymbol{\alpha})|\Psi\rangle\langle\Psi|V(\boldsymbol{\alpha})^\dagger ||_1^2 \right], \quad (1)$$

where the testing distribution $\mathcal{P}$ is a probability distribution over (pure) $n$-qubit states $|\Psi\rangle$ and the factor of 1/4 ensures $0 \leq R_{\mathcal{P}}(\boldsymbol{\alpha}) \leq 1$.

A learner will not have access to the full testing ensemble $\mathcal{P}$ and so cannot evaluate the cost in Eq. (1). Instead, it is typically assumed that the learner has access to a training data set consisting of input-output pairs of pure $n$-qubit states,

$$\mathcal{D}_{\mathcal{Q}}(N) = \left\{ \left( \left|\Psi^{(j)}\right\rangle, U\left|\Psi^{(j)}\right\rangle \right) \right\}_{j=1}^N, \quad (2)$$

where the $N$ input states are drawn independently from a training distribution $\mathcal{Q}$. Equipped with such training data, the learner may evaluate the *training cost*

$$C_{\mathcal{D}_{\mathcal{Q}}(N)}(\boldsymbol{\alpha}) = \frac{1}{4N} \sum_{j=1}^N \left\| U\left|\Psi^{(j)}\right\rangle\left\langle\Psi^{(j)}\right|U^\dagger \right.$$
$$\left. - V(\boldsymbol{\alpha})\left|\Psi^{(j)}\right\rangle\left\langle\Psi^{(j)}\right|V(\boldsymbol{\alpha})^\dagger \right\|_1^2. \quad (3)$$

We note that this cost can be rewritten in terms of the average fidelity as

$$C_{\mathcal{D}_{\mathcal{Q}}(N)}(\boldsymbol{\alpha}) = 1 - \frac{1}{N} \sum_{j=1}^N \left| \left\langle\Psi^{(j)}\right| V(\boldsymbol{\alpha})^\dagger U \left|\Psi^{(j)}\right\rangle \right|^2 \quad (4)$$

and thus can be efficiently computed using a Loschmidt echo[14] or swap test circuit[43,44]. The hope is that by training the parameters $\boldsymbol{\alpha}$ of the QNN to minimize the training cost $C_{\mathcal{D}_{\mathcal{Q}}(N)}(\boldsymbol{\alpha})$ one will also achieve small risk $R_{\mathcal{P}}(\boldsymbol{\alpha})$.

However, whether such a strategy is successful crucially depends on whether the training cost $C_{\mathcal{D}_{\mathcal{Q}}(N)}(\boldsymbol{\alpha})$ is indeed a good proxy for the expected cost $R_{\mathcal{P}}(\boldsymbol{\alpha})$. This is exactly the question of *generalization*: Does good performance on the training data imply good performance on (previously unseen) testing data?

In statistical learning theory, answers to this question are given in terms of *generalization bounds*. These are bounds on the generalization error, which is typically taken to be the difference between expected risk and training cost, i.e.,

$$\mathrm{gen}_{\mathcal{P}, \mathcal{D}_{\mathcal{Q}}(N)}\left(\boldsymbol{\alpha}_{\mathrm{opt}}\right) := R_{\mathcal{P}}\left(\boldsymbol{\alpha}_{\mathrm{opt}}\right) - C_{\mathcal{D}_{\mathcal{Q}}(N)}\left(\boldsymbol{\alpha}_{\mathrm{opt}}\right). \quad (5)$$

Usually, such bounds are proved under an i.i.d. assumption on training and testing. That is, they are based on the assumptions (a) that the training examples are drawn independently from a training distribution $\mathcal{Q}$ and (b) that the training and testing distributions coincide, $\mathcal{Q} = \mathcal{P}$. In this case, we speak of *in-distribution generalization*.

In this paper, we consider *out-of-distribution generalization* where we drop assumption (b) by allowing $\mathcal{Q} \neq \mathcal{P}$. Borrowing classical machine learning terminology, one can also regard this as a scenario of dataset shift[45], or more specifically covariate shift[46,47], which is often addressed using transfer learning techniques[48,49]. We formulate our results for a broad class of ensembles called *locally scrambled* ensembles. In loose terms, locally scrambled ensembles of states can be thought of as ensembles of states that are at least locally random. Throughout, we use the terms 'distribution' and 'ensemble' interchangeably. More formally, locally scrambled ensembles are defined as follows.

**Definition 1.** (Locally scrambled ensembles). An ensemble of *n*-qubit unitaries is called *locally scrambled* if it is invariant under pre-processing by tensor products of arbitrary local unitaries. That is, a unitary ensemble $\mathcal{U}_{LS}$ is locally scrambled iff for $U \sim \mathcal{U}_{LS}$ and for any fixed $U_1, \ldots, U_n \in \mathcal{U}(\mathbb{C}^2)$ also $U(\otimes_{i=1}^{n} U_i) \sim \mathcal{U}_{LS}$. Here and elsewhere, the "~" notation means that the random variable on the left has the distribution on the right as its law. For instance, $U \sim \mathcal{U}_{LS}$ means that the random unitary $U$ is drawn from the distribution $\mathcal{U}_{LS}$. Accordingly, an ensemble $\mathcal{S}_{LS}$ of *n*-qubit quantum states is locally scrambled if it is of the form $\mathcal{S}_{LS} = \mathcal{U}_{LS}|0\rangle^{\otimes n}$ for some locally scrambled unitary ensemble $\mathcal{U}_{LS}$. We use $\mathcal{U}|0\rangle^{\otimes n}$ to denote the ensemble of states generated by drawing unitaries from $\mathcal{U}$ and applying them to the *n*-qubit all-zero state $|0\rangle^{\otimes n}$. We denote the classes of locally scrambled ensembles of unitaries and states as $\mathbb{U}_{LS}$ and $\mathbb{S}_{LS}$, respectively.

In fact, our results hold for a slightly broader class of ensembles where we only require that the ensemble agrees with a locally scrambled one up to and including its (complex) second moments. That is, more informally, the average over the ensemble agrees with those of a locally scrambled ensemble over all functions of $U$ that contain at most two copies of $U$. We will denote these broader classes of unitary and state ensembles, which we formally define in Supplementary Note 1, as $\mathbb{U}_{LS}^{(2)}$ and $\mathbb{S}_{LS}^{(2)}$, respectively.

In our results, we suppose that both the testing and training ensembles are such ensembles, i.e., $\mathcal{P} \in \mathbb{S}_{LS}^{(2)}$ and $\mathcal{Q} \in \mathbb{S}_{LS}^{(2)}$. However, as $\mathbb{S}_{LS}^{(2)}$ captures a variety of different possible ensembles, $\mathcal{P}$ and $\mathcal{Q}$ can be ensembles containing very different sorts of states. In particular, as detailed further in Supplementary Note 1, the following are important examples of ensembles in $\mathbb{S}_{LS}^{(2)}$:

- $\mathcal{S}_{Haar_1^{\otimes n}}$ - Products of Haar-random single-qubit states.
- $\mathcal{S}_{Stab_1^{\otimes n}}$ - Products of random single-qubit stabilizer states.
- $\mathcal{S}_{Haar_k^{\otimes n/k}}$ - Products of Haar-random *k*-qubit states.
- $\mathcal{S}_{Haar_n}$ - Haar-random *n*-qubit states.
- $\mathcal{S}_{2design}$ - A 2-design on *n*-qubit states.
- $\mathcal{S}_{RandCirc}^{\mathcal{A}_k}$ - The output states of random quantum circuits. (Here $\mathcal{A}_k$ denotes the *k*-local *n*-qubit quantum circuit architecture from which the random circuit is constructed.)

These examples highlight that the class of locally scrambled ensembles includes both ensembles that consist solely of product states and ensembles composed mostly of highly entangled states. We can use this to our advantage to construct more efficient machine learning strategies.

Typically the learner will be interested in learning the action of a unitary on a wide class of input states including both entangled and unentangled states. For example, they might be interested in learning the action of a unitary on all states that can be efficiently prepared on a quantum computer using a polynomial-depth hardware-efficient layered ansatz. Thus in general the expected risk should be evaluated over distributions such as $\mathcal{S}_{Haar_n}$, $\mathcal{S}_{2design}$ or $\mathcal{S}_{RandCirc}^{\mathcal{A}_k}$ (for $k \geq 2$) which cover a large proportion of the total Hilbert space.

In classical machine learning, one often thinks of the training data as given. However, in the context of learning or compiling quantum unitary dynamics (as sketched in Fig. 1), one in practice needs either to

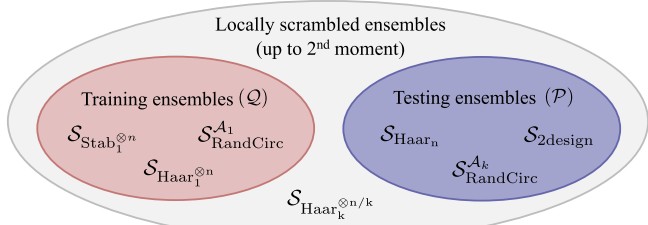

**Fig. 2 | Locally scrambled ensembles.** Venn diagram showing how the class of ensembles that are locally scrambled up to the second moment, $\mathbb{S}_{LS}^{(2)}$, divides naturally into training ensembles and testing ensembles. For the formal definitions of each of the ensembles referenced see Supplementary Note 1.

prepare the training states on a quantum computer or in an experimental setup, or to be able to efficiently simulate them classically. Thus, it is desirable to train on states that can be prepared using simple circuits, i.e., those that are short depth, low-entangling or require only simple gates. This is especially important in the NISQ era due to noise-induced barren plateaus[50] or other noise-related issues[51]. Therefore, as random stabilizer states and random product states can be prepared using only a single layer of single-qubit gates, it makes practical sense to train using the ensembles $\mathcal{S}_{Haar_1^{\otimes n}}$ or $\mathcal{S}_{Stab_1^{\otimes n}}$.

In this manner the class of ensembles that are locally scrambled to the second moment, $\mathbb{S}_{LS}^{(2)}$, divides naturally into sub-classes of ensembles that give rise to training sets and testing sets. We sketch this in Fig. 2.

## Analytical results

Having set up our framework, we now present our analytical results. First, we show that all locally scrambled ensembles lead to closely related expected risks for unitary learning. Second, we use this observation to lift in-distribution generalization to out-of-distribution generalization when using a QNN to learn an unknown unitary from quantum data. For the formal proofs see Supplementary Note 2.

We first show a close connection between the risks for unitary learning arising from any locally scrambled ensembles. More precisely, we show that they can be upper and lower bounded in terms of the expected risk over the Haar distribution in our main technical result:

**Lemma 1.** For any $\mathcal{Q} \in \mathbb{S}_{LS}^{(2)}$ and any parameter setting $\boldsymbol{\alpha}$,

$$\frac{1}{2}R_{\mathcal{S}_{Haar_n}}(\boldsymbol{\alpha}) \leq \frac{d}{d+1}R_{\mathcal{Q}}(\boldsymbol{\alpha}) \leq R_{\mathcal{S}_{Haar_n}}(\boldsymbol{\alpha}), \tag{6}$$

where $d = 2^n$ is the dimension of the target unitary $U$ being learned.

This result establishes that learning over any locally scrambled distribution is effectively equivalent (up to a constant multiplicative factor) to learning over the uniform distribution over the entire Hilbert space. We note that the factor of 1/2 in the lower bound emerges from the structure of our proof, and for typical cases we expect the relation between the costs to be tighter still. We explore this numerically in Supplementary Note 3 for the special case of training on random product states, i.e. $\mathcal{Q} = \mathcal{S}_{Haar_1^{\otimes n}}$.

A direct consequence of Lemma 1 is that the risks arising from any two locally scrambled ensembles are related as follows.

**Theorem 1.** (Equivalence of locally scrambled ensembles for comparing unitaries). Let $\mathcal{P} \in \mathbb{S}_{LS}^{(2)}$ and $\mathcal{Q} \in \mathbb{S}_{LS}^{(2)}$, then for any parameter setting $\boldsymbol{\alpha}$,

$$\frac{1}{2}R_{\mathcal{Q}}(\boldsymbol{\alpha}) \leq R_{\mathcal{P}}(\boldsymbol{\alpha}) \leq 2R_{\mathcal{Q}}(\boldsymbol{\alpha}). \tag{7}$$

Theorem 1 establishes an equivalence (up to a constant multiplicative factor) between all locally scrambled testing distributions for the task of learning an unknown unitary on average. In particular, even simple locally scrambled ensembles, such as tensor products of Haar-random single-qubit states or of random single-qubit stabilizer states, are for this purpose effectively equivalent to seemingly more complex locally scrambled ensembles. The latter include the output states of random quantum circuits or, indeed, globally Haar-random states.

Theorem 1 gives rise to a general template for lifting in-distribution generalization bounds for QNNs to out-of-distribution generalization guarantees in unitary learning. This is captured by the following corollary:

**Corollary 1.** (Locally scrambled out-of-distribution generalization from in-distribution generalization). Let $\mathcal{P} \in \mathbb{S}_{\text{LS}}^{(2)}$ and $\mathcal{Q} \in \mathbb{S}_{\text{LS}}^{(2)}$. Let $U$ be an unknown $n$-qubit unitary. Let $V(\boldsymbol{\alpha})$ be an $n$-qubit unitary QNN that is trained using training data $\mathcal{D}_{\mathcal{Q}}(N)$ containing $N$ input-output pairs, with inputs drawn from the ensemble $\mathcal{Q}$. Then, for any parameter setting $\boldsymbol{\alpha}$,

$$R_{\mathcal{P}}(\boldsymbol{\alpha}) \leq 2 \left( C_{\mathcal{D}_{\mathcal{Q}}(N)}(\boldsymbol{\alpha}) + \text{gen}_{\mathcal{Q}, \mathcal{D}_{\mathcal{Q}}(N)}(\boldsymbol{\alpha}) \right). \quad (8)$$

Thus, when training using training data $\mathcal{D}_{\mathcal{Q}}(N)$, the out-of-distribution risk $R_{\mathcal{P}}(\boldsymbol{\alpha}_{\text{opt}})$ of the optimized parameters $\boldsymbol{\alpha}_{\text{opt}}$ after training is controlled in terms of the optimized training cost $C_{\mathcal{D}_{\mathcal{Q}}(N)}(\boldsymbol{\alpha}_{\text{opt}})$ and the in-distribution generalization error $\text{gen}_{\mathcal{Q}, \mathcal{D}_{\mathcal{Q}}(N)}(\boldsymbol{\alpha}_{\text{opt}})$. We can now bound the in-distribution generalization error using already known QML in-distribution generalization bounds[11–23] (or, indeed, any such bounds that are derived in the future). We point out that our results up to this point do not require any assumptions on the QNN architecture underlying $V(\boldsymbol{\alpha})$, except for overall unitarity. As a concrete example of guarantees that can be obtained this way, we combine Corollary 1 with an in-distribution generalization bound established in[20] to prove:

**Corollary 2.** (Locally scrambled out-of-distribution generalization for QNNs). Let $\mathcal{P} \in \mathbb{S}_{\text{LS}}^{(2)}$ and $\mathcal{Q} \in \mathbb{S}_{\text{LS}}^{(2)}$. Let $U$ be an unknown $n$-qubit unitary. Let $V(\boldsymbol{\alpha})$ be an $n$-qubit unitary QNN with $T$ parameterized local gates. When trained with the cost $C_{\mathcal{D}_{\mathcal{Q}}(N)}$ using training data $\mathcal{D}_{\mathcal{Q}}(N)$, the out-of-distribution risk w.r.t. $\mathcal{P}$ of the parameter setting $\boldsymbol{\alpha}_{\text{opt}}$ after training satisfies

$$R_{\mathcal{P}}(\boldsymbol{\alpha}_{\text{opt}}) \leq 2 C_{\mathcal{D}_{\mathcal{Q}}(N)}(\boldsymbol{\alpha}_{\text{opt}}) + \mathcal{O}\left( \sqrt{\frac{T \log(T)}{N}} \right), \quad (9)$$

with high probability over the choice of training data of size $N$ according to $\mathcal{Q}$.

The out-of-distribution generalization guarantee of Corollary 2 is particularly interesting if the training data is drawn from a distribution composed only of products of single-qubit Haar-random or random stabilizer states, i.e. $\mathcal{Q} = \mathcal{S}_{\text{Haar}_1^{\otimes n}}$ or $\mathcal{Q} = \mathcal{S}_{\text{Stab}_1^{\otimes n}}$, but the testing data is drawn from more complex distributions such as the Haar ensemble or the outputs of random circuits, i.e. $\mathcal{P} = \mathcal{S}_{\text{Haar}_n}$ or $\mathcal{P} = \mathcal{S}_{\text{RandCirc}}$. In this case, Corollary 2 implies that efficiently implementable unitaries can be learned using a small number of simple unentangled training states. More precisely, if $U$ can be approximated via a QNN with $poly(n)$ trainable local gates, then only $poly(n)$ unique product training states suffice to learn the action of $U$ on the Haar distribution, i.e. across the entire Hilbert space.

To understand why out-of-distribution generalization is possible, recall that any state is linearly spanned by $n$-qubit Pauli observables $P \in \{I, X, Y, Z\}^{\otimes n}$, and each Pauli observable $P$ can be written as a linear combination of product states $|s\rangle \langle s| = \bigotimes_{i=1}^{n} |s_i\rangle \langle s_i|$, where $s_i \in \{0, 1, +, -, y+, y-\}$. These two facts imply that for any state $|\phi\rangle \langle \phi|$, there exists coefficients $\alpha_s$, such that $|\phi\rangle \langle \phi| = \sum_s \alpha_s |s\rangle \langle s|$.

Hence, if we exactly know $U|s\rangle \langle s|U^{\dagger}$ for all $6^n$ product states $|s\rangle \langle s|$, then we can figure out $U|\phi\rangle \langle \phi|U^{\dagger}$ for any state $|\phi\rangle \langle \phi|$ by linear combination. However, this requires an exponential number of product states in the training data. In our prior work[20], we show that one only needs $poly(n)$ training product states to approximately know $U|s\rangle \langle s|U^{\dagger}$ for most of the $6^n$ product states, assuming $U$ is efficiently implementable. The key insight in this work is that one can predict $U|\phi\rangle \langle \phi|U^{\dagger}$ as long as the coefficients $\alpha_s$ in $|\phi\rangle \langle \phi| = \sum_s \alpha_s |s\rangle \langle s|$ are sufficiently random and spread out across the $6^n$ product states. We make this condition precise by defining locally scrambled ensembles and proving that the action of $U$ on a state sampled from any such ensemble can be predicted. In Supplementary Note 2, we further discuss the role that linearity plays in our results.

We can immediately extend our results for out-of-distribution to local variants of costs. Such local costs are essential to avoid cost-function-dependent barren plateaus[52] when training a shallow QNN. As a concrete example, when taking $\mathcal{S}_{\text{Haar}_1^{\otimes n}}$ as the training ensemble, we can consider the local training cost

$$C_{\text{Prod},N}^{L}(\boldsymbol{\alpha}) = 1 - \frac{1}{N} \sum_{j=1}^{N} \text{Tr}\left[ \left| \Psi_{\text{Prod}}^{(j)} \right\rangle \left\langle \Psi_{\text{Prod}}^{(j)} \right| U^{\dagger} V(\boldsymbol{\alpha}) H_{L}^{(j)} V(\boldsymbol{\alpha})^{\dagger} U \right], \quad (10)$$

where $\left| \Psi_{\text{Prod}}^{(j)} \right\rangle = \bigotimes_{i=1}^{n} |\psi_i^{(j)}\rangle \sim \mathcal{S}_{\text{Haar}_1^{\otimes n}}$ for all $j$ and we have introduced the local measurement operator $H_{L}^{(j)} = \frac{1}{n} \sum_{i=1}^{n} |\psi_i^{(j)}\rangle \langle \psi_i^{(j)}| \otimes \mathbb{1}_{\bar{i}}$. This local cost is faithful to its global variant for product state training in the sense that it vanishes under the same conditions[30], but crucially, in contrast to the global case, may be trainable[52]. In Supplementary Note 2, we prove a version of Corollary 2 when training on the local cost from Eq. (10). Specifically we find:

**Corollary 3.** (Locally scrambled out-of-distribution generalization for QNNs via a local cost). Let $\mathcal{P} \in \mathbb{S}_{\text{LS}}^{(2)}$ and let $U$ be an unknown $n$-qubit unitary. Let $V(\boldsymbol{\alpha})$ be an $n$-qubit unitary QNN with $T$ parameterized local gates. When trained with the cost $C_{\text{Prod},N}^{L}$, the out-of-distribution risk w.r.t. $\mathcal{P}$ of the parameter setting $\boldsymbol{\alpha}_{\text{opt}}$ after training satisfies

$$R_{\mathcal{P}}(\boldsymbol{\alpha}_{\text{opt}}) \leq 2n C_{\text{Prod},N}^{L}(\boldsymbol{\alpha}_{\text{opt}}) + \mathcal{O}\left( n \sqrt{\frac{T \log(T)}{N}} \right), \quad (11)$$

with high probability over the choice of training data of size $N$.

Clearly, analogous local variants of the training cost can be defined whenever the respective ensemble has a tensor product structure (such as $\mathcal{S}_{\text{Stab}_1^{\otimes n}}$). However, if the training data is highly entangled, constructing such local costs in this manner is not possible. Thus, this is another important consequence of our results: The ability to train solely on product state inputs makes it straightforward to generate the local costs that are necessary for efficient training.

The results presented thus far concern the number of unique training states required for generalization, but in practice multiple copies of each training state will be needed for successful training. As $\mathcal{O}(1/\epsilon^2)$ shots are required to evaluate a cost to precision $\epsilon$ and since for gradient based training methods one needs to evaluate the partial derivative of the cost with respect to each of the $T$ trainable parameters, one would expect to need on the order of $\mathcal{O}(TM_{\text{opt}}/\epsilon^2)$ copies of each of the $N$ input states and output states to reduce the cost to $\epsilon$. Here $M_{\text{opt}}$ is the number of optimization steps. Classical shadow tomography[53–55] provides a way towards a copy complexity bound that is independent of the number of optimization steps. Namely, exploiting covering number bounds for the space of pure output states of polynomial-size quantum circuits (compare refs. [4],[20]), polynomial-size classical shadows can be used to perform tomography among such states. In the case of an efficiently implementable target unitary $U$ and QNN $V(\boldsymbol{\alpha})$ that both admit a circuit representation with $T \in$

$\mathcal{O}(poly(n))$ local gates, $\mathcal{O}(T\log(T/\epsilon)/\epsilon^2) \le \tilde{\mathcal{O}}(poly(n)/\epsilon^2)$ copies of each of the input states $|\Psi^{(j)}\rangle$ and output states $|\Phi^{(j)}\rangle$ suffice to approximately evaluate the cost (both the global and local variants) and its partial derivatives arbitrarily often.

## Numerical results

Here we provide numerical evidence to support our analytical results showing that out-of-distribution generalization is possible for the learning of quantum dynamics. We focus on the task of learning the parameters of an unknown target Hamiltonian by studying the evolution of product states under it.

For concreteness, we suppose that the target Hamiltonian is of the form

$$H(\boldsymbol{p},\boldsymbol{q},\boldsymbol{r}) = \sum_{k=1}^{n-1} (Z_k Z_{k+1} + p_k X_k X_{k+1}) + \sum_{k=1}^{n} (q_k X_k + r_k Z_k), \quad (12)$$

with the specific parameter setting $(\boldsymbol{p}^*,\boldsymbol{q}^*,\boldsymbol{r}^*)$ given by $p_k^* = \sin\left(\frac{\pi k}{2n}\right)$ for $1 \le k \le n-1$ and $q_k^* = \sin\left(\frac{\pi k}{n}\right)$, $r_k^* = \cos\left(\frac{\pi k}{n}\right)$ for $1 \le k \le n$. The learning is performed by comparing the exact evolution under $e^{-iH(\boldsymbol{p}^*,\boldsymbol{q}^*,\boldsymbol{r}^*)t}$ to a Trotterized ansatz. Specifically, we use an $L$ layered ansatz $V_L(\boldsymbol{p},\boldsymbol{q},\boldsymbol{r}) := (U_{\Delta t}(\boldsymbol{p},\boldsymbol{q},\boldsymbol{r}))^L$ where $U_{\Delta t}$ is a second order Trotterization of $e^{-iH(\boldsymbol{p},\boldsymbol{q},\boldsymbol{r})\Delta t}$. That is,

$$U_{\Delta t}(\boldsymbol{p},\boldsymbol{q},\boldsymbol{r}) = e^{-iH_A(\boldsymbol{r})\Delta t/2} e^{-iH_B(\boldsymbol{p},\boldsymbol{q})\Delta t} e^{-iH_A(\boldsymbol{r})\Delta t/2} \quad (13)$$

where the Hamiltonians $H_A(\boldsymbol{r}) := \sum_{k=1}^{n-1} Z_k Z_{k+1} + \sum_{k=1}^{n} r_k Z_k$ and $H_B(\boldsymbol{p},\boldsymbol{q}) := \sum_{k=1}^{n-1} p_k X_k X_{k+1} + \sum_{k=1}^{n} q_k X_k$ contain only commuting terms and so can be readily exponentiated.

We attempt to learn the vectors $\boldsymbol{p}^*$, $\boldsymbol{q}^*$, and $\boldsymbol{r}^*$ by comparing $e^{-iH(\boldsymbol{p}^*,\boldsymbol{q}^*,\boldsymbol{r}^*)t}|\psi_j\rangle$ and $V_L(\boldsymbol{p},\boldsymbol{q},\boldsymbol{r})|\psi_j\rangle$ over $N$ random product states $|\psi_j\rangle$. To do so, we use the training data $\mathcal{D}_{\mathcal{Q}}(N)$ with $\mathcal{Q} = \mathcal{S}_{\text{Haar}_1^{\otimes n}}$, and the cost function given in Eq. (4). The learning is performed classically for $n = 4, ..., 12$ and $L = 2, ..., 5$ and we take the total evolution time to be $t = 0.1$. For all values of $n$ we train on two product states, i.e. $N = 2$. We repeated the optimization 5 times in each case and kept the best run. While the small training data size $N = 2$ was sufficient for the model considered here, in Supplementary Note 3 we present a more involved unitary learning setting that requires larger values of $N$.

Figure 3 plots the in-distribution risk and out-of-distribution risk as a function of the final optimized cost function values, $C_{\mathcal{D}_{\mathcal{Q}}(2)}(\boldsymbol{\alpha}_{\text{opt}})$ with $\mathcal{Q} = \mathcal{S}_{\text{Haar}_1^{\otimes n}}$. Here the in-distribution risk is the average prediction error over random product states, i.e. $R_{\mathcal{S}_{\text{Haar}_1^{\otimes n}}}$, and for the out-of-distribution testing we chose to compute the risk over the global Haar distribution, i.e. $R_{\mathcal{S}_{\text{Haar}_n}}$. These risks can be evaluated analytically using Supplementary Lemma 3 and Supplementary Eqs. (6) and (7). The linear correlation between the cost function and both $R_{\mathcal{S}_{\text{Haar}_1^{\otimes n}}}$ and $R_{\mathcal{S}_{\text{Haar}_n}}$ demonstrates that both in-distribution and out-of-distribution generalization have been successfully achieved.

Next, we perform noisy simulations to assess the performance of learning the parameters of the Hamiltonian in Eq. (12) in two situations: (i) the training is performed on random product states and (ii) the training data is prepared with deep quantum circuits. We expect that the presence of noise will have a different impact depending on the amount of noise that is accumulated during the preparation of the training states.

Our simulations used a realistic noise model based on gate-set tomography on the IBM Ourense superconducting qubit device[56] but with the experimentally obtained error rates reduced by a factor of 20 to make the difference in training more pronounced. The training set is constructed from just two states (either product states or those prepared with a linear depth hardware efficient circuits).

The optimizer is a version of the gradient-free Nelder–Mead method[57]. The cost function in Eq. (4) is computed with an increasing

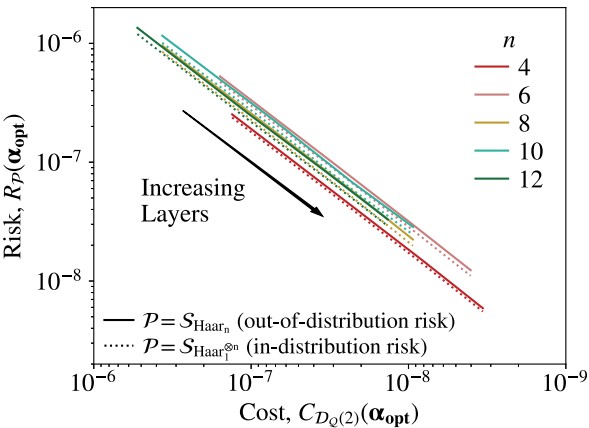

**Fig. 3 | Out-of-Distribution Generalization for Hamiltonian Learning.** Here we present our results from learning the Hamiltonian specified in Eq. (12) by training on only 2 product states. As the number of layers $L$ in the ansatz is increased the obtainable cost function value decreases. We plot the correlation between the optimized cost $C_{\mathcal{D}_{\mathcal{Q}}(2)}(\boldsymbol{\alpha}_{\text{opt}})$ with $\mathcal{Q} = \mathcal{S}_{\text{Haar}_1^{\otimes n}}$, and the (in-distribution) risk over product states, $R_{\mathcal{S}_{\text{Haar}_1^{\otimes n}}}$, and out-of-distribution) risk over the Haar measure, $R_{\mathcal{S}_{\text{Haar}_n}}$. The lines indicate the joined values for $L = 2, 3, 4, 5$ for the different values of $n$ indicated in the legend.

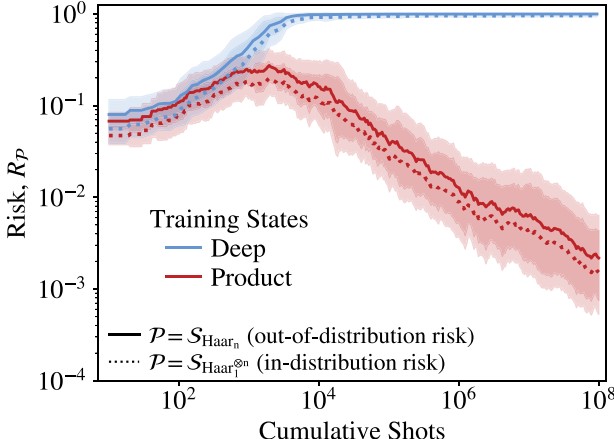

**Fig. 4 | Training in the presence of noise.** The cost function is optimized for two types of training data: (i) product states (red lines) and (ii) states prepared with deep circuits (blue lines). We performed 20 independent optimizations, each time initializing the optimization differently and selecting a different random training set. The shaded region represents the standard deviation of all 20 runs. Dotted (solid) lines represent in-distribution (out-of-distribution) risk.

number of shots, starting with 10 shots per cost function evaluation. That number is increased by 50% once the optimizer detects a lack of progress within a specified number of iterations. This optimization procedure is sensitive to flatness of the cost function landscape: The flatter the landscape, the more shots are needed to resolve it and find a minimizing direction.

Figure 4 shows the results of the training procedure performed on an $n = 6$ qubit system. Here, we train the $L = 2$ ansatz for $V_L(\boldsymbol{p},\boldsymbol{q},\boldsymbol{r})$ and consider total evolution time $t = 0.1$. The optimization is repeated 20 times, each time starting with different random initial point $(\boldsymbol{p}_0,\boldsymbol{q}_0,\boldsymbol{r}_0)$. Red (blue) lines indicate the risk obtained for product (deep circuit) training states as a function of total number of shots.

Training with product states is successful: once the number of shots per cost function evaluation is large enough (total shots above $10^3$), the optimizer detects the downhill direction and the in-distribution risk is gradually decreased, eventually reaching $10^{-3}$. The out-of-distribution risk closely follows the in-distribution risk

proving that generalization can be achieved with product training states under realistic noise and finite shot budget conditions. In contrast, the training set built with deep circuits fails to produce successful training for all 20 optimization runs. Even in the limit of very large number of shots, both in-distribution and out-of-distribution risks remain large. This proof-of-principle numerical experiment shows that our out-of-distribution generalization guarantees can make training and learning feasible in noisier scenarios than otherwise viable.

## Discussion

Our work establishes that for learning unitaries, QNNs trained on quantum data enjoy out-of-distribution generalization between some physically relevant distributions if the training data size is roughly the number of trainable gates. The class of locally scrambled distributions that our results hold for fall naturally into sub-classes of training ensembles and testing ensembles, characterized by their practicality and generality, respectively. The simplest possible training ensemble in this context are products of stabilizer states. Our results show that training on this easy to experimentally prepare and easy to classically simulate ensemble generalizes to the uniform Haar ensemble of states, as well as to practically motivated ensembles such as the output of random circuits. Thus, somewhat surprisingly, we have shown the action of quantum unitaries can be predicted on a wide class of highly entangled states, having only observed their action on relatively few unentangled states.

These results have implications for the practicality of learning quantum dynamics. We are particularly intrigued by the possibility of using quantum hardware or experimental systems to characterize unknown dynamics of quantum experimental systems. This could be done by coherently interacting a quantum system with a quantum computer, or alternatively could be conducted in a more conventional experimental setup. We stress for the latter, the experimental setup may not be equipped with a complete gate set, and so our proof that learning can be done using only products of random single qubit states, which require only simple single-qubit gates to prepare, is particularly important.

We are also interested in the potential of these results to ease the classical compilation of local short-time evolutions into shorter depth circuits[30] and circuits of a particular desired structure[34,35]. Since low-entangling unitaries and product states may be classically simulated using tensor network methods, our results show that the compilation of such unitaries may be performed entirely classically. This could be used to develop more effective methods for dynamical simulation or to learn more efficient pulse sequences for noise resilient gate implementations.

An immediate extension of our results would be to investigate whether our proof techniques can be used to more efficiently evaluate Haar integrals, or more generally to relate averages over different locally scrambled ensembles in other settings. For example, one might explore whether they could be used in a DQC1 (deterministic quantum computation with 1 clean qubit) setting where one inputs a maximally mixed state[58]. Alternatively, one might investigate whether they could be used to bound the frame potential of an ensemble, an important quantity for evaluating the randomness of a distribution that has links with quantifying chaotic behavior[59].

In this paper, we have focused on the learning of quantum dynamics, in particular the learning of unitaries, using locally scrambled distributions. Given recent progress on different quantum channel learning questions[60–70], it is natural to ask whether out-of-distribution generalization is possible for other QML tasks such as learning quantum channels or, more generally, for performing classification tasks such as classifying phases of matter[20,71,72]. (We note that our proof techniques extend beyond unitary dynamics to doubly stochastic quantum channels, which can be understood as affine combinations of unitary channels [ref. 73, Theorem 1].) It would further be valuable to investigate whether out-of-distribution generalization is viable for other classes of distributions. Such results, if obtainable, would again have important implications for the practicality of QML on near term hardware and restricted experimental settings.

Our approach to out-of-distribution generalization does not rely on specific learning algorithms, nor transfer learning techniques, as is often the case in the classical literature[45–49]. Rather, we establish generalization guarantees that apply to a specific QML task (learning quantum dynamics) with data coming from a specific class of distributions (locally scrambled ensembles). That is, we show that in this context, out-of-distribution generalization is essentially automatic. In the classical ML literature, a similar-in-spirit focus on properties of the class of distributions of interest can for example be seen in the concepts of invariance[74,75] and variation[76] of features, but the nature of these properties is still quite different from the ones that we consider. Nevertheless, we hope that combining such perspectives from classical ML theory with physics-informed choices of distributions, as in our case, will lead to a better understanding of out-of-distribution generalization.

## Methods

In this section, we give an overview over the proof strategy leading to our central analytical result contained in Lemma 1. At a high level, our proof boils down to rewriting $R_{\mathcal{S}_{\mathrm{Haar}_n}}(\boldsymbol{\alpha})$ and $R_{\mathcal{Q}}(\boldsymbol{\alpha})$ with $\mathcal{Q}$ locally scrambled into forms which are comparable by known and newly derived inequalities.

First, we recast the Haar risk $R_{\mathcal{S}_{\mathrm{Haar}_n}}(\boldsymbol{\alpha})$ into an average over Pauli products and upper bound it by a risk over local stabilizer states. To do so, we rewrite the Haar risk by recalling the relationship between the (Haar) average gate fidelity between two unitaries $U$ and $V$ and the Hilbert–Schmidt inner product[77],

$$
\begin{aligned}
R_{\mathcal{S}_{\mathrm{Haar}_n}}(\boldsymbol{\alpha}) &= \mathbb{E}_{|\Psi\rangle \sim \mathcal{S}_{\mathrm{Haar}_n}}\left[1 - |\langle\Psi|U^{\dagger}V(\boldsymbol{\alpha})|\Psi\rangle|^2\right] \\
&= \frac{d}{d+1}\left(1 - \frac{1}{d^2}|\mathrm{Tr}[U^{\dagger}V(\boldsymbol{\alpha})]|^2\right).
\end{aligned}
\tag{14}
$$

Next, we use the Pauli basis expansion of the swap operator to write the Haar risk as an average over Pauli operators. That is, as shown explicitly in Supplementary Lemma 1, we use

$$
\mathrm{SWAP} = \sum_{P \in \{1,X,Y,Z\}^{\otimes n}} P \otimes P
\tag{15}
$$

to show that

$$
|\mathrm{Tr}[U^{\dagger}V]|^2 = \frac{1}{d}\sum_{P \in \{1,X,Y,Z\}^{\otimes n}} \mathrm{Tr}[PU^{\dagger}VPV^{\dagger}U].
\tag{16}
$$

This gives an expression for the Haar risk $R_{\mathcal{S}_{\mathrm{Haar}_n}}(\boldsymbol{\alpha})$ in terms of an average over Pauli products. This average over Pauli observables can then be upper bounded by an average over products of stabilizer states by introducing a spectral decomposition, as detailed in Supplementary Lemma 2 and Supplementary Corollary 1. Finally, by the 2-design property of the random single-qubit stabilizer states, we can rewrite this upper bound in terms of a local Haar average,

$$
\begin{aligned}
&\frac{d+1}{d}R_{\mathcal{S}_{\mathrm{Haar}_n}}(\boldsymbol{\alpha}) \leq 2(1-\chi) \quad \text{where}, \\
&\chi = \mathbb{E}_{\otimes_{i=1}^{n}|\psi_i\rangle \sim \mathrm{Haar}_1^{\otimes n}}\left[\left|\left(\bigotimes_{i=1}^{n}\langle\psi_i|\right)\tilde{U}^{\dagger}W\tilde{U}\left(\bigotimes_{i=1}^{n}|\psi_i\rangle\right)\right|^2\right].
\end{aligned}
\tag{17}
$$

The latter can then be related to $R_{\mathcal{Q}}(\boldsymbol{\alpha})$ because $\mathcal{Q}$ is locally scrambled, which then leads to the first inequality in Lemma 1. Here the choice to

bound by $\text{Haar}_1^{\otimes n}$ specifically hints towards our final result that a unitary can be learnt over the Haar average from product state training data.

Second, we recast the generic locally scrambled risk $R_{\mathcal{Q}}$ into a sum of locally scrambled expectation values over different partitions of the system. Specifically, using a well known expression for the complex second moment of the single-qubit Haar measure (see, e.g., Eq. (2.26) in ref. 59),

$$\mathbb{E}_{|\psi\rangle \sim \text{Haar}_1}\left[\big||\psi\rangle\langle\psi|^{\otimes 2}\big|\right] = \frac{\mathbb{1} \otimes \mathbb{1} + \text{SWAP}}{6}, \qquad (18)$$

we find that

$$R_{\mathcal{Q}}(\boldsymbol{\alpha}) = 1 - \frac{1}{6^n}\sum_{A \subseteq \{1,\ldots,n\}} \mathbb{E}_{\tilde{U} \sim \tilde{\mathcal{U}}}\left\|\text{Tr}_{A^c}\left[\tilde{U}^\dagger U^\dagger V \tilde{U}\right]\right\|_F^2, \qquad (19)$$

where $\tilde{U} \sim \tilde{\mathcal{U}}$ is drawn from the locally scrambled unitary ensemble $\tilde{\mathcal{U}}$ with $\mathcal{Q} = \tilde{U}|0\rangle^{\otimes n}$. See Supplementary Lemma 3 for more details. From here, we can use matrix-analytic inequalities to show a lower bound on the Frobenius norm of a partial trace of a matrix in terms of the absolute value of the trace of the original matrix. Plugging this lower bound into the explicit expression for $R_{\mathcal{Q}}(\boldsymbol{\alpha})$ translates exactly to the second inequality in Lemma 1.

## Data availability
The data generated and analyzed during the current study are available from the authors upon request.

## Code availability
Further implementation details are available from the authors upon request.

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

## Acknowledgements

We thank Marco Cerezo for helpful conversations. We thank the reviewers at Nature Communications for their valuable feedback. M.C.C. was supported by the TopMath Graduate Center of the TUM Graduate School at the Technical University of Munich, Germany, the TopMath Program at the Elite Network of Bavaria, by a doctoral scholarship of the German Academic Scholarship Foundation (Studienstiftung des deutschen Volkes), by the BMWK (PlanQK), and by a DAAD PRIME Fellowship. N.E. was supported by the U.S. DOE, Department of Energy Computational Science Graduate Fellowship under Award Number DE-SC0020347. H.-Y.H. is supported by a Google PhD Fellowship. P.J.C. and A.T.S. acknowledge initial support from the Los Alamos National Laboratory (LANL) ASC Beyond Moore's Law project. Research presented in this paper (A.T.S.) was supported by the Laboratory Directed Research and Development (LDRD) program of Los Alamos National Laboratory under project number 20210116DR. L.C. acknowledges support from LDRD program of LANL under project number 20230049DR. L.C. and P.J.C. were also supported by the U.S. DOE, Office of Science, Office of Advanced Scientific Computing Research, under the Accelerated Research in Quantum Computing (ARQC) program. Z.H. acknowledges support from the LANL Mark Kac Fellowship and from the Sandoz Family Foundation-Monique de Meuron program for Academic Promotion.

## Author contributions

The project was conceived by M.C.C., H.-Y.H., A.T.S., L.C., P.J.C., and Z.H. Theoretical results were proved by M.C.C., H.-Y.H., and Z.H. Numerical implementations were performed by N.E., J.G., and L.C. The manuscript was written by M.C.C., H.-Y.H., N.E., J.G., A.T.S., L.C., P.J.C., and Z.H.

## Funding

## Competing interests

The authors declare no competing interests.
