## [Peer Review File · Nature Communications]

Out-of-distribution generalization for learning quantum dynamicsREVIEWER COMMENTS

Reviewer #1 (Remarks to the Author):

In this paper, the authors studied the ability of out-of-distribution generalization of a quantum machine learning model when it is applied to learn unknown quantum dynamics. The authors considered the task of learning an unknown unitary U , given a dataset $\mathcal{D}_Q(N) = \{(|\psi_i\rangle, U|\psi_i\rangle)\}_{i=1}^N$, where input states are randomly sampled from an ensemble Q . More specifically, the training ensemble Q and testing ensemble P can be different and are both locally scrambled ensembles, which cover a big family of random states ranging from product states to highly entangled states. First, the authors established the equivalence of expected risks among locally scrambled ensembles, showing that expected risks for different locally scrambled ensembles are approximately the same up to a constant multiplicative factor. Then the authors calculated the out-of-distribution generalization bound between empirical training cost and expected risk on different locally scrambled ensembles. The theoretical results are supported by numerical experiments on learning unknown local Hamiltonian dynamics. Both in-distribution and out-of-distribution expected risk shows good agreement with the empirical training cost.

Overall, the theoretical results of the out-of-distribution (OOD) generalization of quantum machine learning are very intriguing. Especially the OOD bound shows it is possible to use easy-to-prepare quantum states for learning unitary dynamics and predicts the unitary evolution of other non-trivial quantum states, which may have a big impact on the applications of quantum machine learning in the near future. The article is written clearly, and the theoretical derivations are arranged in a way that readers can follow without extra effort. I think this article meets the high standard of Nature Communications, and I recommend its publication.

Here are some of my minor questions:

1. In the experiments (Sec. C), the authors considered learning of a local Hamiltonian dynamics with $T = 0.1$ and showed that the expected risk and empirical cost have perfect correlations for both in-distribution and out-of-distribution tests. I wonder if one enlarges the evolution time T or changes the underlying Hamiltonian to a non-local Hamiltonian such that the unitary evolution will generate more entanglement, such as volume-law entanglement, will the out-of-distribution test still hold? In other words, if unitary dynamics can generate large entanglement, will the learning on random product states will still be efficient? From the theorem, it seems the generalization bound will still hold, but it would be good to confirm it numerically.
2. In the experiments (Sec. C), the authors used two product states as the training dataset. This is a very small training set. I wonder what is the reason for choosing such a small training set. If one does the same training with more training data, say 100 training data pairs, will the result change?
3. In Fig.4, the authors evaluated the expected risk \mathcal{R}_P , which is the expectation value over the ensemble P . In practice, how do the authors evaluate \mathcal{R}_P ? Is it calculated analytically or evaluated with many samples from P ?
4. At the beginning of page 6, the authors change the notation from $H(a, b, c)$ to $H(p, q, r)$ and $U(p, q, r)$. Maybe it is better to keep one notation since $H(a, b, c)$ is only used once in the definition.

Reviewer #2 (Remarks to the Author):

In this reviewed paper, the authors explored out-of-distribution generalization for the task of learning unitary dynamics. Specifically, the authors rigorously proved that the expected risks of all locally scrambled distributions for unitary learning are closely related. There are two key insights behind this conclusion. For the first point, an exponential number of computational basis product states are enough to capture all parameters of a quantum unitary, and any state in the Hilbert space can be written as a linear superposition of these basis product states. From this perspective, the expected risks of all locally scrambled ensembles can be related to the expected risk of all basis product states. Therefore, the quantum neural network trained well on product states can also generalize well to the entire Hilbert space. For the second point, in previous work, the authors have shown that only a polynomial number of product states is required to know the action of a unitary on all basis product states if the target unitary is efficiently implementable. This conclusion makes out-of-distribution generalization possible when the training and testing distributions are both locally scrambled. Such results could be of considerable importance for the practical task of learning unitary in the NISQ era since quantum neural networks trained on product states that can be efficiently prepared on quantum devices can generalize well to the entire Hilbert space.

To numerically demonstrate out-of-distribution generalization for learning quantum unitary dynamics, the authors consider the task of learning the unitary evolution of a target Hamiltonian with unknown parameters. A multiple-layered Trotterization of the target Hamiltonian serves as the ansatz to learn the unitary dynamics with only two product training states. The numerical results displayed in the manuscript suggest that both in-distribution generalization and out-of-distribution generalization are achieved in this task. The authors also perform noisy simulations to study how the noise accumulated during the preparation of the training states affects training. The numerical results suggest noisy training is feasible on product states prepared using only single-qubit gates, but not feasible on the training data prepared with deep quantum circuits.

The results in the paper are scientifically sound, and the paper is very well presented. We believe this work is an important contribution, since to the best of our knowledge it is the first work to study out-of-distribution generalization in quantum machine learning in a both analytical and numerical fashion. We would recommend the publication of the manuscript in Nature Communications, provided that the authors satisfactorily address the following comments.

1) For the numerical simulation in the manuscript, the authors consider the task of learning the unitary evolution of a target Hamiltonian with unknown parameters. The ansatz to learn the unitary dynamics is a multiple layered Trotterization of the target Hamiltonian. The training data consists of only two product states. From the numerical results in the manuscript, we observed that both in-distribution generalization and out-of-distribution generalization are achieved even for large system sizes. This fact may contradict the conclusion of Corollary 2 in the manuscript, which claims that the number of training states required for training needs to scale at least linearly with the number of parameters in the ansatz. In our opinion, choosing the multiple-layered Trotterization of the target Hamiltonian as the ansatz greatly reduces the difficulty of learning the unitary evolution. Could it be possible that a few training states are enough to learn the unknown parameters under such a hypothesis set, leaving not much room for overfitting the training states?

If the authors agree with us, we suggest they choose a different ansatz, such as a parameterized quantum circuit. The numerical results for such a choice may match well with the conclusion of Corollary 2 and thus can support the analytical results of this work.

2) It could be that the good out-of-distribution generalization performance is merely due to the linearity of learning unitaries, since if the quantum neural network can perform well on two states, then will perform well on any linear combination of them. But this will not be true for a classification task or other tasks that lack linearity. Would the authors remark on this point?

3) The notation n is a bit confusing. In the section "framework", n is the number of qubits for the

quantum unitary, but becomes the size of the target Hamiltonian in the section “numerical results”. Also, T denotes the number of parameterized gates of a quantum neural network in the section “analytical results”, but becomes the evolution time in the section “numerical results”.

Reviewer #3 (Remarks to the Author):

"However, in practice one may only have access to a limited type of training data, and yet be interested in making accurate predictions for a wider class of inputs. This is particularly an issue in the Noisy Intermediate Scale Quantum (NISQ) era [25], when hardware limitations and circuit errors limit the quantum training data that can be efficiently prepared."

I find this to be vague. The authors mean that hardware limitations would enable a non-statistically representative subset of training data. Akin to a situation which arose with e.g. Google QAOA experiment: the instances that were considered were in the low-density subset of the possible instances, and hence the data was on the cusp of an abrupt fall off in the types of problem instances which could be minimised at all [Quantum 5, 532 (2021)].

The authors write, "we initiate a study" — please check journal policies on claims of novelty. I did not check NCOM but a claim to novelty can be challenged later whereas it is never absolutely possible to prove that this does not exist somewhere in the literature. It is a caveat but most journals insist on no such claims (yet the work would not be published if it was not novel).

The authors insist on putting a soft bracket after figure declarations, then do things like this: (Fig. 2c)) I won't judge the contents of the paper on this but...Please don't do that. I'd drop the soft right bracket. Maybe the journal will change this in the typesetting process but to me it looks unusual.

Now I have to say that the introduction has two figures, not either of which I found particularly illuminating. The authors write, " • That is, rather intriguingly,
• 72 we show that one can learn the action of such a unitary
• 73 on a broad spread of highly entangled states having only
• 74 studied its action on a limited number of product states."

Is this not a consequence of linearity? For example, say I learn the action $|1\rangle \rightarrow |1'\rangle$ and $|2\rangle \rightarrow |2'\rangle$ do I not know the action of any $a|1\rangle + b|2\rangle$ by linearity?

Finally the authors write, " • Namely, we find that the out-of-
• 79 distribution generalization error (in particular, the aver-
• 80 age learning error over Haar random states) nearly per-
• 81 fectly correlates with the in-distribution generalization
• 82 error (over product states) and the training cost. "

I would personally better quantify this statement and spend the entirety of the introduction explaining this. There is ample redundancy in the abstract. But it is up to the authors what they do here, I am only going to judge the paper based on the results and not these style points.

II Results.

The authors write $U \in U(C^{2^n})$ ignoring the tensor product structure of $[C^2]^{\otimes n}$ and thereby forgetting qubit labelings. Evidently if $U \in [C^2]^{\otimes n}$ what they said is also true, yet I imagine the qubits have labels and the tensor product structure is used.

In Eq (1), $|\Psi$ is not defined in the main body of the text before its use here; the symbol did appear in the caption of Fig 1 however. It is defined in Eq (1) as being sampled from P , yet in figure 1 it is drawn from Q . In the figure, $P=Q$ in the testing ensemble. Still, personally I would be happier if the authors

would considering explaining the variables and the terms in the text, and not just in the figure's caption.

Personally I would have formulated this differently, you want to learn U by V so you sample over states Ψ . So eqn 1, which you can't evaluate fully, it can be a trace difference squared, which is minimal iff $V = U$. Maybe I am missing something here but it seems that is the ultimate goal.

I would remove "crucially" and loose nothing on page 3.

And finally the authors don't follow their own bracket convention, line 221 "(as sketched in Fig. 2)"

What is your training ansatz or what is the QNN structure?

The authors write "It is natural to ask 558 whether out-of-distribution generalization is possible for 559 other QML tasks such as learning quantum channels [64]" which references a footnote. However, there is work on learning channels [Physical Review A 106 (3), 032409].

Overall the results are about generalisation error yet the paper (and I understand this is not an easy thing to do) is not entirely clear at all points. It fails to clarify certain things and in an order that I think is optimal. The main point as I see it, there is a QNN to learn some unitary and this is done training only on product states, which can be easily prepared. Certainly the preparation of input states for machine learning is a challenge, we even considered that in [Physical Review A 102, 012415].

Overall I would say that the work can be polished and appear in the scientific record. It baffles me why there are so many worthless citations that are tangential to the main results yet many results (not just mine) inside the QML literature are not mentioned and compared against. I would remove all references that are not related to the subject. And reduce the number of references to better match the topic.

Response to Comments, Criticisms, and Suggestions of Reviewer #1

Dear Reviewer,

Thank you for your kind review of our manuscript and for the positive feedback with constructive criticism. In particular, in response to your comments, we have included additional numerical experiments, added clarifications on how the testing risks in Fig. 4 were evaluated analytically, and partially updated the notation in Section II C. Please see below for our detailed responses to your comments and questions, and compare the attached redline version, in which changes to the manuscript are highlighted in either blue or red.

“In this paper, the authors studied the ability of out-of-distribution generalization of a quantum machine learning model when it is applied to learn unknown quantum dynamics. The authors considered the task of learning an unknown unitary U , given a dataset $\mathcal{D}_{\mathcal{Q}}(N) = \{(|\psi_i\rangle, U|\psi_i\rangle)\}_{i=1}^N$, where input states are randomly sampled from an ensemble. More specifically, the training ensemble \mathcal{Q} and testing ensemble \mathcal{P} can be different and are both locally scrambled ensembles, which cover a big family of random states ranging from product states to highly entangled states. First, the authors established the equivalence of expected risks among locally scrambled ensembles, showing that expected risks for different locally scrambled ensembles are approximately the same up to a constant multiplicative factor. Then the authors calculated the out-of-distribution generalization bound between empirical training cost and expected risk on different locally scrambled ensembles. The theoretical results are supported by numerical experiments on learning unknown local Hamiltonian dynamics. Both in-distribution and out-of-distribution expected risk shows good agreement with the empirical training cost.”

We consider this summary of our work to be accurate.

“Overall, the theoretical results of the out-of-distribution (OOD) generalization of quantum machine learning are very intriguing. Especially the OOD bound shows it is possible to use easy-to-prepare quantum states for learning unitary dynamics and predicts the unitary evolution of other non-trivial quantum states, which may have a big impact on the applications of quantum machine learning in the near future. The article is written clearly, and the theoretical derivations are arranged in a way that readers can follow without extra effort. I think this article meets the high standard of Nature Communications, and I recommend its publication.”

We are grateful for the positive evaluation of our work and for the appreciation of our efforts in making the presentation as clear as possible.

“Here are some of my minor questions:”

Thank you for your questions, these are addressed next.

“1. In the experiments (Sec. C), the authors considered learning of a local Hamiltonian dynamics with $T = 0.1$ and showed that the expected risk and empirical cost have perfect correlations for both in-distribution and out-of-distribution tests. I wonder if one enlarges the evolution time T or changes the underlying Hamiltonian to a non-local Hamiltonian such that the unitary evolution will generate more entanglement, such as volume-law entanglement, will the out-of-distribution test still hold? In other words, if unitary dynamics can generate large entanglement, will the learning on random product states still be efficient? From the theorem, it seems the generalization bound will still hold, but it would be good to confirm it numerically.”

Thank you for this interesting question. We agree that our theoretical results still apply, even for larger evolution times and non-local Hamiltonians. Concretely, according to Corollary 2, even in those cases, a training data set of size growing approximately linearly with the number of trainable gates will (with high probability) lead to good out-of-distribution testing, assuming good training.

Based on your suggestion, we performed additional numerical experiments to take into account evolutions that generate more entanglement, see Appendix C2. There, we investigate a QNN/parametrized quantum circuit ansatz modelling a so-called fast scrambler, which can induce large amounts of entanglement. In this case, the analogue to the evolution time in our Hamiltonian learning experiments is the number of time steps (layers) in the ansatz, which we assume to be the same as in the unknown fast scrambler target unitary. In our new experiments, we find that our theoretical generalization guarantees also apply to this more involved setting. However, we also see that

the more involved the ansatz and the target – in the sense of larger number of time steps –, the larger the training data size required for good generalization, getting closer to our theoretical upper bound. Still, the required training data size remains moderate and the type of training data remains the same, that is, we still train on (relatively few) random product states. From this, we conclude that our out-of-distribution guarantees still hold even in the case of unitaries that can generate significant entanglement, but that here the training data size cannot be as small as for the short-time Hamiltonian dynamics considered in the main text.

“2. In the experiments (Sec. C), the authors used two product states as the training dataset. This is a very small training set. I wonder what is the reason for choosing such a small training set. If one does the same training with more training data, say 100 training data pairs, will the result change?”

Thank you for this question. Our motivation for working with a training data set consisting of only two product states was to demonstrate good generalization with as small a training data set as possible in our numerical experiments. However, we agree that the question of how the results change for larger data sets is natural. Based on your suggestion, we have reevaluated the data that we had already collected for larger data sets. There, we did not see any significant changes in the results.

However, following up on your question, we have investigated the relevance of the training data size in more detail in the additional numerical experiments in Appendix C 2. There, we see that the more involved the ansatz, the larger the training data size required for good generalization, getting closer to our theoretical upper bound. While the required training data size remains moderate even in this more complicated scenario, this indicates that the extremely small training data size of 2 sufficient for the numerical results in the main text was specific to that (relatively simple) model. In particular, we observed for the more involved scrambler model that increasing the training data size improves the generalization behavior and thereby the chances of successfully learning the unknown unitary. We have added a corresponding comment to the main text, in which we refer to Appendix C 2 for our results on the more involved ansatz.

“3. In Fig.4, the authors evaluated the expected risk $R_{\mathcal{P}}$, which is the expectation value over the ensemble \mathcal{P} . In practice, how do the authors evaluate $R_{\mathcal{P}}$? Is it calculated analytically or evaluated with many samples from \mathcal{P} ?”

Thank you for pointing out to us that this was not clear from our presentation. We evaluate $R_{\mathcal{P}}$ analytically, using Lemma B.3. We have added a corresponding sentence to Section II C.

“4. At the beginning of page 6, the authors change the notation from $H(\mathbf{a}, \mathbf{b}, \mathbf{c})$ to $H(\mathbf{p}, \mathbf{q}, \mathbf{r})$ and $U(\mathbf{p}, \mathbf{q}, \mathbf{r})$. Maybe it is better to keep one notation since $H(\mathbf{a}, \mathbf{b}, \mathbf{c})$ is only used once in the definition.”

Thank you for pointing this issue with our notation out to us. We were using $(\mathbf{a}, \mathbf{b}, \mathbf{c})$ for the specific target parameter setting used to generate the training data, whereas we took $(\mathbf{p}, \mathbf{q}, \mathbf{r})$ to stand for the trainable parameters. However, we agree that this change in notation can be confusing. We have therefore adapted the notation, please see the redline version for details.

We hope that this satisfactorily answers your questions. If you have further questions, please let us know!

Response to Comments, Criticisms, and Suggestions of Reviewer #2

Dear Reviewer,

Thank you for your in-depth review of our manuscript and for the constructive suggestions for improvement. Based on your comments, we have added further numerical experiments beyond the ones included in the original submission, included additional clarification regarding the role of linearity, and partially updated our notation in Section II C. Please see below for our detailed responses to your comments and questions, and compare the attached redline version, in which changes to the manuscript are highlighted in either blue or red.

“In this reviewed paper, the authors explored out-of-distribution generalization for the task of learning unitary dynamics. Specifically, the authors rigorously proved that the expected risks of all locally scrambled distributions for unitary learning are closely related. There are two key insights behind this conclusion. For the first point, an exponential number of computational basis product states are enough to capture all parameters of a quantum unitary, and any state in the Hilbert space can be written as a linear superposition of these basis product states. From this perspective, the expected risks of all locally scrambled ensembles can be related to the expected risk of all basis product states. Therefore, the quantum neural network trained well on product states can also generalize well to the entire Hilbert space. For the second point, in previous work, the authors have shown that only a polynomial number of product states is required to know the action of a unitary on all basis product states if the target unitary is efficiently implementable. This conclusion makes out-of-distribution generalization possible when the training and testing distributions are both locally scrambled. Such results could be of considerable importance for the practical task of learning unitary in the NISQ era since quantum neural networks trained on product states that can be efficiently prepared on quantum devices can generalize well to the entire Hilbert space.”

We consider this summary of the theoretical part of our work to be accurate, with one small exception: Our out-of-distribution generalization results apply to (e.g.) tensor products of random single-qubit stabilizer states, not to computational basis product states. We explain this in more detail in our reply to comment 2) below.

“To numerically demonstrate out-of-distribution generalization for learning quantum unitary dynamics, the authors consider the task of learning the unitary evolution of a target Hamiltonian with unknown parameters. A multiple-layered Trotterization of the target Hamiltonian serves as the ansatz to learn the unitary dynamics with only two product training states. The numerical results displayed in the manuscript suggest that both in-distribution generalization and out-of-distribution generalization are achieved in this task. The authors also perform noisy simulations to study how the noise accumulated during the preparation of the training states affects training. The numerical results suggest noisy training is feasible on product states prepared using only single-qubit gates, but not feasible on the training data prepared with deep quantum circuits.”

We consider this summary of the numerical part of our work to be accurate.

“The results in the paper are scientifically sound, and the paper is very well presented. We believe this work is an important contribution, since to the best of our knowledge it is the first work to study out-of-distribution generalization in quantum machine learning in a both analytical and numerical fashion. We would recommend the publication of the manuscript in Nature Communications, provided that the authors satisfactorily address the following comments.”

Thank you for the positive evaluation of our work. Below, we address your comments.

“1) For the numerical simulation in the manuscript, the authors consider the task of learning the unitary evolution of a target Hamiltonian with unknown parameters. The ansatz to learn the unitary dynamics is a multiple layered Trotterization of the target Hamiltonian. The training data consists of only two product states. From the numerical results in the manuscript, we observed that both in-distribution generalization and out-of-distribution generalization are achieved even for large system sizes. This fact may contradict the conclusion of Corollary 2 in the manuscript, which claims that the number of training states required for training needs to scale at least linearly with the number of parameters in the ansatz. In our opinion, choosing the multiple-layered Trotterization of the target Hamiltonian as the ansatz greatly reduces the difficulty of learning the unitary evolution. Could it be possible that a few training states are enough to learn the unknown parameters under such a hypothesis set, leaving not much room for overfitting the training states? If the authors agree with us, we suggest they choose a different ansatz, such as a parameterized

quantum circuit. The numerical results for such a choice may match well with the conclusion of Corollary 2 and thus can support the analytical results of this work.”

Thank you for this observation. We agree that the good generalization from only two training data states in our numerical experiments is stronger than what is guaranteed by our analytical results. However, as our analytical results give upper bounds on the sufficient training data size (rather than lower bounds on the necessary training data size), these numerical experiments do not contradict our analytical results.

Nevertheless, we agree that the gap between our theoretical guarantee and the sufficient training data size observed in our numerical experiment is striking. To investigate whether this difference persists for a more involved QNN ansatz, we performed additional numerical experiments, which are presented in Appendix C 2. There, we investigate a QNN/parametrized quantum circuit ansatz modelling a so-called fast scrambler, which can induce large amounts of entanglement. In our experiments, we find that our theoretical generalization guarantees also apply to this more involved ansatz. However, we see that the more involved the ansatz, the larger the training data size required for good generalization, getting closer to our theoretical upper bound. While the required training data size remains moderate even in this more complicated scenario, this indicates that the extremely small training data size of 2 sufficient for the numerical results in the main text was specific to that (relatively simple) model. In particular, we observed for the more involved scrambler model that increasing the training data size improves the generalization behavior and thereby the chances of successfully learning the unknown unitary. We have added a corresponding comment to the main text, in which we refer to Appendix C 2 for our results on the more involved ansatz.

“2) It could be that the good out-of-distribution generalization performance is merely due to the linearity of learning unitaries, since if the quantum neural network can perform well on two states, then will perform well on any linear combination of them. But this will not be true for a classification task or other tasks that lack linearity. Would the authors remark on this point?”

Thank you for this insightful question. We agree that linearity is important in enabling our out-of-distribution generalization, which we attempted to explain in the discussion between Corollary 2 and Corollary 3. The underlying thought behind your questions seems to be that, as long as the training states span the space on which you wish to learn the action of the target unitary, it ought to be possible to train on those states and by linearity be guaranteed to have learned on the entire space. However, to the best of our understanding, this line of argument is not sufficient to explain out-of-distribution generalization. The random ensembles of states also have to be “well-behaved” to ensure efficient generalization from a manageable number of training states.

One way of highlighting this subtlety would be to note that even an exponential number of computational basis states cannot be used to learn an unknown unitary using a cost formulated in terms of the 1-norm distance between the guess output and true output (or equivalently the fidelity between the guess and true outputs). Namely, computational basis states do not allow to learn relative phases. In response to you pointing out this important conceptual question, we have added a variant of this short explanation together with a concrete illustrating example as Appendix B 3.

A second point that we want to emphasize: An argument based on linearity places no guarantees on how many training states are required/sufficient for convergence. The argument that ‘as long as the training states span the space on which you wish to learn the action of the target unitary on, it ought to be possible to train on those states and by linearity be guaranteed to have learned on the entire space’ crucially only applies if you train on an exponentially large training ensemble. In general, how many states are required/sufficient to ensure good generalization will depend on the types of states in the training ensemble. For example, when computing averages of Hermitian operators $\langle\langle\psi|H|\psi\rangle\rangle_\psi$, the average agrees for Haar-random n -qubit states and for tensor products of Haar-random single-qubit states since both form a 1-design; however, for a generic choice of H , the global Haar average converges exponentially faster than the average over random product states. A similar phenomenon may hold for the case of unitary learning: Intuitively, more random product states may be required to learn a unitary than fully random states. In our work, we combine Theorem 1 with the recent in-distribution generalization bounds of [20] and thereby show that the worst-case training data requirements for random product states cannot be significantly worse than those for fully random states.

In summary: Linearity alone does not trivially imply out-of-distribution generalization, one has to identify suitable ensembles of states and then quantify the training data requirements.

Finally, we stress that our results here are specific to the costs and risks that we consider for learning unitary processes. Therefore, our theoretical results do not guarantee out-of-distribution generalization for other, potentially non-linear costs/risk, as would have to be considered for classification tasks. Whether we can expect out-of-distribution generalization between these ensembles also for non-linear loss functions is an open question (compare also our discussion in Section III).

“3) The notation n is a bit confusing. In the section “framework”, n is the number of qubits for the quantum unitary,

but becomes the size of the target Hamiltonian in the section “numerical results”. Also, T denotes the number of parameterized gates of a quantum neural network in the section “analytical results”, but becomes the evolution time in the section “numerical results”.

Thank you for pointing this out. We agree that the use of T once for the number of parameterized gates of a QNN and later as evolution time is an undesirable doubling in notation. Therefore, we have renamed the evolution time to t . However, we do not see an issue with how n is used in our notation, as the number of qubits in the framework section corresponds to the size of the Hamiltonian in Section II C. Therefore, we have not changed that part of our notation.

We hope that this satisfactorily addresses your comments and that you now recommend publication. If you have further concerns, please let us know!

Response to Comments, Criticisms, and Suggestions of Reviewer #3

Dear Reviewer,

Thank you for your review of our manuscript and for your attention to detail. Based on your comments, we have removed Fig. 1, included additional relevant references, removed some only indirectly relevant references, included additional clarification regarding the role of linearity, and implemented improvements to both notation and wording. We believe that these changes improved the manuscript and thank you for your suggestions! Please see below for our detailed responses to your comments and questions, and compare the attached redline version, in which changes to the manuscript are highlighted in either blue or red. We hope that the change in the numbering of the references (caused by following your suggestions on including and removing references) does not cause confusion.

“‘However, in practice one may only have access to a limited type of training data, and yet be interested in making accurate predictions for a wider class of inputs. This is particularly an issue in the Noisy Intermediary Scale Quantum (NISQ) era [25], when hardware limitations and circuit errors limit the quantum training data that can be efficiently prepared.’ I find this to be vague. The authors mean that hardware limitations would enable a non-statistically representative subset of training data. Akin to a situation which arose with e.g. Google QAOA experiment: the instances that were considered were in the low-density subset of the possible instances, and hence the data was on the cusp of an abrupt fall off in the types of problem instances which could be minimised at all [Quantum 5, 532 (2021)].”

Thank you for this criticism and for the reference that you provided. However, the two sentences that you quoted express a simpler NISQ restriction than the involved considerations in [Quantum 5, 532 (2021)]. We merely mean that a large part of state space is only accessible via (noiseless) deep quantum circuits, and is thus out of reach for limited depth of (noisy) NISQ circuits. Therefore, if only NISQ devices are available, states that require many circuit layers to be prepared are not suitable quantum training data states in practice. In response to your criticism, we have reformulated the respective two sentences, please see the redline version for the exact reformulation.

“The authors write, “we initiate a study” — please check journal policies on claims of novelty. I did not check NCOM but a claim to novelty can be challenged later whereas it is never absolutely possible to prove that this does not exist somewhere in the literature. It is a caveat but most journals insist on no such claims (yet the work would not be published if it was not novel).”

Thank you for this suggestion. We have rephrased this, now we avoid the formulation “we initiate a study”.

“The authors insist on putting a soft bracket after figure declarations, then do things like this: (Fig. 2c) I won’t judge the contents of the paper on this but...Please don’t do that. I’d drop the soft right bracket. Maybe the journal will change this in the typesetting process but to me it looks unusual.”

Thank you for pointing this out. The double-closing bracket appeared because the label of the panel in the figure was “c”, which was then followed by a closing bracket. We agree that it looks unusual and have corrected it.

“Now I have to say that the introduction has two figures, not either of which I found particularly illuminating.”

Thank you for this criticism. We would be grateful for any suggestions regarding which aspects of Figs. 1 and 2 you would urge us to improve. After some consideration, we have decided to remove Fig. 1. We are still of the opinion that Fig. 2 gives an intuitive depiction of different scenarios in which learning quantum dynamics can be relevant, therefore we have decided to keep this figure.

“The authors write, ‘That is, rather intriguingly, we show that one can learn the action of such a unitary on a broad spread of highly entangled states having only studied its action on a limited number of product states.’ Is this not a consequence of linearity? For example, say I learn the action $|1\rangle \mapsto |1'\rangle$ and $|2\rangle \mapsto |2'\rangle$ do I not know the action of any $a|1\rangle + b|2\rangle$ by linearity?”

Thank you for this insightful question! Thank you for this insightful question. We agree that linearity is important in enabling our out-of-distribution generalization, which we attempted to explain in the discussion between Corollary 2 and Corollary 3. The underlying thought behind your questions seems to be that, as long as the training states span

the space on which you wish to learn the action of the target unitary, it ought to be possible to train on those states and by linearity be guaranteed to have learned on the entire space. However, to the best of our understanding, this line of argument is not sufficient to explain out-of-distribution generalization. The random ensembles of states also have to be “well-behaved” to ensure efficient generalization from a manageable number of training states.

One way of highlighting this subtlety would be to note that even an exponential number of computational basis states cannot be used to learn an unknown unitary using a cost formulated in terms of the 1-norm distance between the guess output and true output (or equivalently the fidelity between the guess and true outputs). Namely, computational basis states do not allow to learn relative phases. In response to you pointing out this important conceptual question, we have added a variant of this short explanation together with a concrete illustrating example as Appendix B 3.

A second point that we want to emphasize: An argument based on linearity places no guarantees on how many training states are required/sufficient for convergence. The argument that ‘as long as the training states span the space on which you wish to learn the action of the target unitary on, it ought to be possible to train on those states and by linearity be guaranteed to have learned on the entire space’ crucially only applies if you train on an exponentially large training ensemble. In general, how many states are required/sufficient to ensure good generalization will depend on the types of states in the training ensemble. For example, when computing averages of Hermitian operators $\langle\langle\psi|H|\psi\rangle\rangle_\psi$, the average agrees for Haar-random n -qubit states and for tensor products of Haar-random single-qubit states since both form a 1-design; however, for a generic choice of H , the global Haar average converges exponentially faster than the average over random product states. A similar phenomenon may hold for the case of unitary learning: Intuitively, more random product states may be required to learn a unitary than fully random states. In our work, we combine Theorem 1 with the recent in-distribution generalization bounds of [20] and thereby show that the worst-case training data requirements for random product states cannot be significantly worse than those for fully random states.

In summary: Linearity alone does not trivially imply out-of-distribution generalization, one has to identify suitable ensembles of states and then quantify the training data requirements.

Finally, we stress that our results here are specific to the costs and risks that we consider for learning unitary processes. Therefore, our theoretical results do not guarantee out-of-distribution generalization for other, potentially non-linear costs/risk, as would have to be considered for classification tasks. Whether we can expect out-of-distribution generalization between these ensembles also for non-linear loss functions is an open question (compare also our discussion in Section III).

“Finally the authors write, ‘Namely, we find that the out-of-distribution generalization error (in particular, the average learning error over Haar random states) nearly perfectly correlates with the in-distribution generalization error (over product states) and the training cost.’ I would personally better quantify this statement and spend the entirety of the introduction explaining this. There is ample redundancy in the abstract. But it is up to the authors what they do here, I am only going to judge the paper based on the results and not these style points.”

We thank the reviewer for sharing this sentiment. In response, we now stress this point in the preceding paragraph of the introduction, to highlight that this is not just a numerical observation but the core of our theoretical contributions (compare Theorem 1). We have also modified the sentence quoted and added a sentence after it, with an additional explanation of what we found in our numerical experiments. Finally, we have enriched the discussion of our numerical findings in Section II C with further quantitative details of the precise correlation found.

“The authors write $U \in \mathcal{U}(\mathbb{C}^{2^n})$ ignoring the tensor product structure of $(\mathbb{C}^2)^{\otimes n}$ and thereby forgetting qubit labelings. Evidently if $U \in \mathcal{U}((\mathbb{C}^2)^{\otimes n})$ what they said is also true, yet I imagine the qubits have labels and the tensor product structure is used.”

Thank you for pointing this out. We were indeed using the isometric isomorphism $(\mathbb{C}^2)^{\otimes n} \cong \mathbb{C}^{2^n}$ throughout the paper, as we pointed out in the first paragraph of Appendix A. However, you are right that making the tensor product structure explicit is the cleaner way of presenting it, so we have adapted the notation accordingly.

“In Eq (1), Ψ is not defined in the main body of the text before its use here; the symbol did appear in the caption of Fig 1 however. It is defined in Eq (1) as being sampled from \mathcal{P} , yet in figure 1 it is drawn from \mathcal{Q} . In the figure, $\mathcal{P} = \mathcal{Q}$ in the testing ensemble. Still, personally I would be happier if the authors would considering explaining the variables and the terms in the text, and not just in the figure’s caption.”

We agree that $|\Psi\rangle$ in Eq. (1) is only defined implicitly as the random object over which the expectation is taken and appreciate the suggestion of making this clearer. To make this more explicit, we modified the half-sentence after Eq. (1). Regarding the comparison to Fig. 1: Eq. (1) defines the expected testing risk, which corresponds to the risk in the lower two boxes in Fig. 1, in which the state is drawn from \mathcal{P} and \mathcal{P}' , respectively.

“Personally I would have formulated this differently, you want to learn U by V so you sample over states Ψ . So eqn 1, which you can’t evaluate fully, it can be a trace difference squared, which is minimal iff $V = U$. Maybe I am missing something here but it seems that is the ultimate goal.”

The ultimate goal is to learn U such that $U \approx V$. However, the question is what error measure to use to judge the similarity between U and V . You could, as the reviewer suggests, take the error measure to be a matrix norm difference between U and V . We take a different approach and suppose one is interested in minimizing the error in the output states of a set of test states evolved under U and V . In general, there are different sets of test states one could use – hence us leaving Eq. (1) general. However, if the states are chosen from the Haar measure then our error measure effectively reduces to the trace distance.

“I would remove ‘crucially’ and lose nothing on page 3.”

Thank you for this suggestion. We have removed the word “crucially” as suggested.

“And finally the authors don’t follow their own bracket convention, line 221 ‘(as sketched in Fig. 2)’”

We have corrected the appearing double-closing brackets.

“What is your training ansatz or what is the QNN structure?”

Thank you for this question. In fact, one strength of our results up to and including Corollary 1 in the main text is that they hold true independently of the chosen ansatz or QNN structure, they apply to any parametrized unitary. The bounds in Corollaries 2 and 3 still hold for a variety of ansätze or QNN structures, as long as they have at most T parametrized local gates (and a fixed quantum circuit structure). Thus, we intentionally do not specify the chosen ansatz or QNN structure when presenting our analytical results. To clarify this point, we have added a corresponding sentence in the paragraph after Corollary 1. For our numerical experiments, naturally, we had to choose ansätze, and we describe those ansätze in Section II C.

“The authors write ‘It is natural to ask whether out-of-distribution generalization is possible for other QML tasks such as learning quantum channels [64]’ which references a footnote. However, there is work on learning channels [Physical Review A 106 (3), 032409].”

In that sentence, we added the footnote [64] to specifically comment on out-of-distribution generalization in quantum channel learning. To the best of our knowledge, there were no works on out-of-distribution generalization in quantum channel learning when we submitted our paper. However, we appreciate your suggestion to include more general references on quantum channel learning. We have added corresponding references, these also include [Physical Review A 106 (3), 032409], which you brought to our attention.

“Overall the results are about generalisation error yet the paper (and I understand this is not an easy thing to do) is not entirely clear at all points. It fails to clarify certain things and in an order that I think is optimal. The main point as I see it, there is a QNN to learn some unitary and this is done training only on product states, which can be easily prepared.”

You have correctly identified our main point, namely that training only on (few) product states is sufficient for unitary learning with an efficiently implementable unitary QNN. We are sorry to hear that you do not consider our presentation clear at all points and would be grateful for any concrete suggestions for improvement. We appreciate that you consider the order in which we present our findings optimal. In an attempt to further clarify our results, we have made some reformulations and additions in the introduction, with the goal of highlighting generalization from product state training as our main point. Please see the redline version of the manuscript for details.

“Certainly the preparation of input states for machine learning is a challenge, we even considered that in [Physical Review A 102, 012415].”

We agree that the preparation of input states for (quantum) machine learning is an important challenge. Thank you for bringing [Physical Review A 102, 012415] to our attention. We have added a citation to it when we discuss potential extensions to classification tasks in Section III.

“Overall I would say that the work can be polished and appear in the scientific record. It baffles me why there are so many worthless citations that are tangential to the main results yet many results (not just mine) inside the QML literature are not mentioned and compared against. I would remove all references that are not related to the subject. And reduce the number of references to better match the topic.”

We are grateful for the evaluation that our work is worthy of publication after some polishing. We hope that the changes that we have implemented constitute such a polishing. In particular, we have included some additional works on QML, especially on quantum channel learning. As we have included all results from the QML literature on generalization in QML that we are aware of already in our initial submission, we would be grateful for concrete suggestions of which additional QML references to include. Based on your criticism, we have also gone through our list of references and removed some that are only indirectly related to our work.

“The authors really reference a lot of things which I think are not directly relevant.”

We appreciate the criticism. Below, we explain why we include the references that you singled out. While not all of them are directly relevant, we cite them to either support statements made in the paper or to provide a broader context. However, we also agree with you that some of the references are not strictly necessary, we have removed the citations of those references.

“Ref 71-80 are all about t -designs.”

We agree that Refs. [83–89] (the numbering was updated because of updates to the manuscript) are all about t -designs. In Example A.5, we cite [83–85] for polynomial bounds on the quantum circuit depth sufficient to obtain an approximate 2-design. In the proof of Corollary B.1, we cite [86–89] for design properties of stabilizer states. As we use these properties either in an example or in a proof, we believe that these citations are appropriate.

To our understanding, Refs. [90–92] are not about t -designs. Rather, we cite them in the context of (minimal) decompositions for 2-qubit gates in Appendix C1. We believe that these citations are appropriate, since they help explain our ansatz.

“I ask the authors to explain what parts of reference 70 are used in this work? Authors should be citing primary sources outside of topic, and primary and relevant sources in top. The topic here is variational circuits.”

In footnote [64], we explain that our proof strategy is in principle not limited to learning unitary dynamics, it can be extended to so-called doubly stochastic quantum channels. (Admittedly, the results obtained for doubly stochastic quantum channels are not as strong as in the unitary case, which is why we do not present these preliminary results in the paper.) Doubly stochastic quantum channels are defined to be completely positive maps that are both trace-preserving and unital. We claim in the footnote that this abstract definition is equivalent to a more concrete representation, namely a quantum channel is doubly stochastic if and only if it can be written as an affine combination of unitary channels. This characterization of doubly stochastic quantum channels was proved in [82, Theorem 1], which is why we cite Ref. [82].

“I ask the authors to explain what part of reference 68 was used in their study.”

As noted in the sentence before our Eq. (B2), we are not the first to prove this equation. Rather, it has been observed previously in Ref. [30], who obtained it as a consequence of results due to [81, 93]. Concretely, our Eq. (B1) follows from [93, Eq. (30)], for which [81] later gave a simplified proof. Given that we are not the first to observe this equality, we considered it appropriate to cite these two (to the best of our knowledge original) references, as well as [30] because the latter reference demonstrated how to translate [93, Eq. (30)] for the HST cost. Based on the reviewer’s suggestion, we have removed the citation of [93]. (In the redline version, it still appears in the list of references, but only because we cite it here in the response.)

“The authors reference ”G. Vidal, Efficient classical simulation of slightly entangled quantum computations, Phys. Rev. Lett. 91, 147902” yet they are implying a different notion of “slightly entangled” in the paper.”

Thank you for pointing this out. Based on your suggestion, we have removed the citation to Vidal’s work to avoid confusion.

“By even the title of 47 I can be sure it is secondary source and should not be cited as primary research ‘S. Bozinovski, Reminder of the first paper on transfer learning in neural networks, 1976, Informatica 44, 10.31449/inf.v44i3.2828 (2020).”

We agree with the reviewer that this is a secondary source. Following the reviewer’s suggestion, we have now removed this citation from the main text.

“Please explain the relevance of reference 42.”

Ref. [43] and [44] are the two of earliest papers that we could identify in which the quantum circuit for a swap test is explicitly described, see [43, Fig. 1] and [44, Section 3]. Given that we mention the swap test circuit as a way of efficiently evaluating our training cost from Eq. (4) on a quantum computer, we consider it appropriate to mention these (to the best of our knowledge original) references proposing the circuit.

“Please explain what was used from reference 41.”

Ref. [42] is widely considered to be the foundational paper for classical computational and statistical learning theory. In particular, the framework of Probably Approximately Correct (PAC) learning, viewed simultaneously from a statistical and a computational perspective, goes back to Ref. [42]. As our theoretical results are formulated in a PAC-like way, we consider it appropriate to reference this seminal paper, even if our results are for quantum learning problems rather than for classical learning.

We hope that this satisfactorily addresses your comments and that you now recommend publication. If you have further concerns, please let us know!

REVIEWERS' COMMENTS

Reviewer #1 (Remarks to the Author):

In the revised version, the authors have effectively addressed the concerns raised in the previous review. They have provided clarification on the notation and evaluation of expected risk and expanded their numerical experiments in Appendix C. The additional experiments demonstrate that the theorem holds when the quantum learning process generates high entanglement. Furthermore, the authors have offered valuable insights regarding the size of the training data. All of my previous questions have been satisfactorily answered, and I believe that the current version is now suitable for publication.

Reviewer #2 (Remarks to the Author):

I went through the revised manuscript, the reports from other referees, and the author's replies to all referees. In this round, the authors have made considerable efforts to improve the manuscript and address all the referees' comments and suggestions. As far as I am concerned, the authors' responses are convincing and satisfying. The revised manuscript has been improved substantially. I am happy to recommend its acceptance in Nature Communications with the current version.

Response to Comments, Criticisms, and Suggestions of Reviewer #1

Dear Reviewer,

Thank you for your positive review of our revised manuscript. Please see below for a detailed response to your review.

“In the revised version, the authors have effectively addressed the concerns raised in the previous review. They have provided clarification on the notation and evaluation of expected risk and expanded their numerical experiments in Appendix C. The additional experiments demonstrate that the theorem holds when the quantum learning process generates high entanglement. Furthermore, the authors have offered valuable insights regarding the size of the training data. All of my previous questions have been satisfactorily answered, and I believe that the current version is now suitable for publication.”

We thank the reviewer for their positive feedback regarding our additions.

We are grateful that you recommended our revised manuscript for publication. Thank you again for your feedback in the previous round, which helped us improve our manuscript significantly.

Response to Comments, Criticisms, and Suggestions of Reviewer #2

Dear Reviewer,

Thank you for your positive review of our revised manuscript. Please see below for a detailed response to your review.

“I went through the revised manuscript, the reports from other referees, and the author’s replies to all referees. In this round, the authors have made considerable efforts to improve the manuscript and address all the referees’ comments and suggestions. As far as I am concerned, the authors’ responses are convincing and satisfying. The revised manuscript has been improved substantially. I am happy to recommend its acceptance in Nature Communications with the current version.”

We thank the reviewer for their positive feedback regarding our additions.

We are grateful that you recommended our revised manuscript for publication. Thank you again for your feedback in the previous round, which helped us improve our manuscript significantly.

Best wishes,
Matthias C. Caro (On behalf of all authors)